# Individual Motivations, Motivational Climate, Enjoyment, and Physical Competence Perceptions in Finnish Team Sport Athletes: A Prospective and Retrospective Study

**DOI:** 10.3390/sports6040165

**Published:** 2018-12-05

**Authors:** Thaís Zanatta, Christoph Rottensteiner, Niilo Konttinen, Marc Lochbaum

**Affiliations:** 1Department of Kinesiology and Sport Management, Texas Tech University, Lubbock, TX 79409, USA; thais.benoit@ttu.edu; 2Research Institute for Olympic Sports, 40700 Jyväskylä, Finland; rottensteiner.christoph@gmail.com (C.R.); niilo.konttinen@kihu.fi (N.K.); 3Department of Education Sciences, Vytautas Magnus University, 44248 Kaunas, Lithuania

**Keywords:** motivation, elite talent development, perceived competence, sport enjoyment, youth sport, sports club

## Abstract

Despite the high rates of participation in sports clubs among Finnish youth, only a few reach elite levels. This study investigated a number of motivational factors, enjoyment, and perceived physical competence perceptions of Finnish youth athletes in their adolescence and then four years later to help understand determinants of elite level attainment. The sample consisted of 824 young athletes born in 1995, who were playing soccer, ice hockey, or basketball in the Finnish sports club system. As youths, participants completed measurements of the perceived task and ego climates, task and ego goal orientations, autonomous and controlled motivations, amotivation, sport enjoyment, and perceived physical competence. Retrospectively, the same participants completed measurements of task, ego, social relatedness, and autonomy supportive climates four years later. All variables were compared to self-reported elite status attainment. Additionally, we examined some demographic characteristics. Prospectively, the self-reported elite athletes (*n* = 79) reported significantly (*p* < 0.05) higher perceptions of a task climate, perceived physical competence, sport enjoyment, and autonomous motivation and a lower level of amotivation compared to nonelite athletes. The meaningfulness (Hedges’ *g*) of the significant differences ranged from small to moderate. Retrospectively, elite athletes indicated significantly (*p* < 0.05) higher perception of a task climate and a social relatedness climate during their sporting career. Hedges’ *g* ranged from moderate to large in meaningfulness. The findings highlighted the importance of focusing on the positive aspects surrounding elite athletes’ perceptions to promote youth athletes’ development, while not discounting the importance of physical size and talent.

## 1. Introduction

Sports science talent development research explores the mechanisms that assist individuals during the process of becoming elite athletes [1,2]. Due to the dynamism of sports settings, multidisciplinary perspectives emphasize the combination of physical and psychological factors as an important aspect of talent development [3]. Certainly, most studies do not investigate all possible physical and psychological combinations. Nevertheless, reviews on this topic do exist, such as a very recent quantitative review by Murr and his colleagues [4] of talent predictors that pull together the importance of physical and psychological factors. For instance, in their systematic review of psychosocial factors in soccer contexts, Gledhill et al. [5] synthesized the associations of 48 psychosocial factors. Some identified factors were intrinsic motivation, social support, sport enjoyment, perceived competence, task/mastery orientation, and an effective learning environment. In addition to Gledhill and colleagues, other researchers [6,7,8] have summarized and continue to examine the importance of similar psychological factors as well as mental skills in elite talent development. The current manuscript builds upon several of the psychosocial factors associated with elite talent development with a large Finnish athlete sample.

In the Finnish society, elite sport is funded by the Ministry of Education and Culture, but different organizations are responsible for coordinating and promoting the development of elite sport [9]. Sport clubs are at the bottom of the Finnish elite sport organization system; however, such organizations play an important role in the earlier stages of athletic development [9]. Currently, Finland has more 10,000 sport clubs, and a substantial number (estimated at 350,000) youth are engaged in sport club activities each year [10]. Although seemingly substantial in number, other reports [11] estimate 49% of girls and 61% of boys aged 7–14 years in Finland to be involved in weekly sports club participation. These percentages dramatically drop as youth get older (i.e., 15–19 years of age) [11]. As an attempt to better understand the association of psychosocial and environmental factors in the Finnish athletes’ development, we resort to two well-established motivational theories in the field of sport psychology, organismic integration theory (OIT) and achievement goal theory (AGT). OIT is one of the minitheories within self-determination theory (SDT) [12]. This study is not an attempt to integrate OIT and AGT constructs, although potential overlaps exist [13]; instead, we adopted a complementary perspective.

OIT is a taxonomy of regulatory styles integral to the range of nonself-determined to self-determined behaviors [12]. From these regulatory styles, the autonomously controlled continuum is formed. The autonomously controlled continuum establishes that autonomous types of motivation, such as intrinsic motivation, derives from oneself and relates to feelings of enjoyment inherent to the activity, whereas controlled types of motivation, such as extrinsic motivation, refer to behaviors coerced by external factors [14]. In the sport domain, OIT research within the SDT framework focuses on identifying the factors that enhance intrinsic motivation and lead to success in elite sport [15]. For instance, social support is considered a primary factor for athletes’ high performance. Research has demonstrated that parents’, peers’, and coaches’ support increase athletes’ autonomous motivation [16,17,18]. Contrarily, rewards viewed as controlling undermined athletes’ intrinsic motivation [17]. 

AGT is a much-researched theory in the sport domain [7,19]. The AGT is centered upon two conceptions of ability, differentiated and undifferentiated, that lead to normative evaluations (ego orientation) or self-referenced evaluations (task orientation), respectively. Individuals with an undifferentiated conception of ability achieve success by focusing on personal improvement, which is known as task orientation. Moreover, individuals adopting a differentiated conception of ability focus on exceeding the performance of others to achieve success; established as an ego orientation [20]. The motivational climate is another element of AGT and refers to the influence of the perceived dominance of one or both of the goal orientations by others, such as coaches [21]. Interactions between goal orientations and motivational climate underline individuals’ state of achievement goal involvement. Although individuals are believed to be inclined towards either task or ego involvement, different circumstances may affect their state of involvement, leading to changes in their perceptions of success [22]. Achievement goal research showed that individuals with different goal orientations present distinctive reasoning for engagement in sport; for example, individuals with higher task orientation demonstrate more intrinsic rationale than individuals with higher ego orientation [19].

In addition to OIT and AGT constructs, we examined sport enjoyment and perceived competence. Both constructs are of great importance within SDT overall and AGT as well as singular constructs. Additionally, enjoyment is termed one of several ‘relevant regulatory processes’ within OIT [12]. Sport enjoyment and perceived competence have gained importance in assisting youth athletes’ development in organized sport [23,24]. Enjoyment associates with participants’ positive sport experience, and thus, it is considered a primary factor in youths’ engagement in organized sports [25]. The same way, perceived competence plays a vital role as a predictor of self-determined types of motivation in the sport context [26,27]. 

In summary, our purpose was to analyze OIT and AGT psychosocial factors with a Finnish athlete sample in two different time periods, with the intent to obtain a better understanding of elite talent development in Finland. Especially based on the current state of the literature, we hypothesized that elite athletes would self-report higher individual self-perceptions, task orientation, perceived competence, sport enjoyment, and autonomous motivation than the Finnish athletes not attaining elite status. Additionally, we hypothesized that the elite athletes would report higher perceptions of the task, socially related, and autonomy-supportive climates compared to the nonelite athletes.

## 2. Materials and Methods

### 2.1. Participants

The data set presented in this study was collected in 2010, or Time 1 (T1), and 2014/15, or Time 2 (T2). Participants initially answering in T1 were followed up in T2 as described below in the procedures. In total, 824 (489 male and 335 female) Finnish youth sport club athletes composed the study’s sample. All participants were born in 1995 and had a valid Finnish sports club license. Only youth in the soccer, ice hockey, or basketball federations were surveyed. As such, the sample sports were soccer (*n* = 431), ice hockey (*n* = 248), and basketball (*n* = 145). At T2, 79 participants (26 male and 53 female) self-identified as having achieved elite athlete status. To the potential participants, we defined elite status as currently or formerly playing professionally in a league that competed internationally, making at least one Finnish Olympic team, or representing Finland in international competitions other than the Olympics (e.g., the World Championships). The rest of the participants were thus categorized as nonelite. Both groups, on average, were very similar based on their self-reported date of birth (birth quartile of elite, 2.45 ± 1.05; nonelite, 2.35 ± 1.09), height (elite, 1.71 ± 0.08 m; nonelite, 1.71 ± 0.08 m), and weight (elite, 61.02 ± 8.40; nonelite, 60.77 ± 9.35).

### 2.2. Measurements

#### 2.2.1. Perceived Motivational Climate in Sport Questionnaire (PMCSQ)

In this study, we used the PMCSQ [28] to examine youth athletes’ perception of a coach-created task-involved and ego-involved climate. Participants answered 24 items regarding task-involved features (e.g., “The coach cared about the players’ development”), and ego-involved features (e.g., “Teammates competed against each other”). The participants answered on a 5-point Likert scale ranging from 1 (strongly disagree) to 5 (strongly agree). Concerning the psychometric properties of the PMCSQ development, Liukkonen [28] reported Cronbach [29] alpha values for scores reported for the task climate scale at 0.86 and 0.84 for the ego climate scale. Liukkonen reported the confirmatory factor analysis also revealed an acceptable level of statistical fit of the scores for both scales.

#### 2.2.2. Perception of Success Questionnaire (POSQ)

The Finnish version of the POSQ [30] measured the young athletes’ dispositional achievement goal orientations. The scale consists of 6 items that measure individuals’ dispositional task orientation and ego orientation. The respondents answered each question based on the statement: “When I am doing sports, I feel most successful when…”. Using a 5-point Likert scale, the participants rated each item ranging from 1 (strongly disagree) to 5 (strongly agree). With regard to the psychometric properties of the Finnish POSQ, Liukkonen and Leskinen [30] reported high validity and reliability of the POSQ based on their participants’ scores. Specifically, they reported Cronbach [29] alpha values of 0.87 for the task and 0.85 for the ego subscales. Additionally, they reported that the confirmatory factor analysis revealed a high structural validity for the POSQ. Continued research with the Finnish POSQ has resulted in the same psychometric properties, e.g., as presented in [31].

#### 2.2.3. Perceived Physical Competence Scale (PPCS)

The participants’ physical competence was examined using the modified Finnish PPCS [32], somewhat akin to items found in Fox and Corbin’s Physical Self-Perception Profile [33]. The Finnish PPCS has ten bipolar items across two subscales: (1) perceived physical performance capacity (7 items) and (2) perceived appearance (3 items). Only the perceived physical performance capacity items were used in this study. Based on a 5-point Osgood scale, the participants rated themselves on specific components compared with those of other players of the same gender and age. The items were strength, speed, agility, flexibility, endurance, movement skills, and courage. The sum of all items determined young athletes’ perceived physical competence. The evidence for the reliability and validity of the PPCS has been provided in previous investigations to some extent in Lintunen’s original work [32] and subsequently in Liukkonen’s [30] study involving 557 14-year-old Finnish football players. Liukkonen reported a Cronbach’s alpha [29] for the summed scores of 0.77 and confirmatory factor analysis with a moderate level of statistical fit.

#### 2.2.4. Sport Enjoyment Scale (SES)

Sport enjoyment was measured using the Finnish version of the SES [28]. The scale comprises four items, and the participants answered on a 5-point Likert scale ranging from 1 (strongly disagree) to 5 (strongly agree). The scale contained questions regarding soccer, ice hockey, and basketball context, for example, “I enjoy basketball training/games.”. The validity and reliability of the SES has been found to be adequate in the Finnish language. For instance, Liukkonen [30] surveyed 557 14-year-old Finnish football players by examining young athletes’ willingness to take up training and games by two single items. He reported high internal consistency of the scores for the scale in both training (Cronbach alpha = 0.90) and game (Cronbach alpha = 0.92) contexts. The exploratory factor analysis with principal axis factorization reported a good level of statistical fit based on the youths’ scores on the two single items in both contexts [28].

#### 2.2.5. Sport Motivation Scale (SMS)

The Finnish version of the SMS [31] was used to assess athletes’ motivation towards organized sports. The scale consists of 28 items and seven subscales, comprising three types of intrinsic motivation (to know, to accomplish things, and to experience stimulation), three types of extrinsic motivation (external introjected and identified regulation), and amotivation. The participants answered the following stem question, “Why do you practice your sport?”, and then were asked to rate each item on a 5-point Likert scale ranging from 1 (strongly disagree) to 5 (strongly agree). The researchers [34] combined the subscales of the SMS into two sum scales: autonomous motivation (intrinsic motivation and identified regulation) and controlled motivation (introjected regulation and external regulation). The mean of both variables was used in the statistical analysis. The Finnish SMS has demonstrated acceptable reliability and validity [31] based on study participants’ scores. In Jaakkola’s first study with the translated SMS [31], involving 461 students aged 15 years, the Cronbach alpha [29] values for the different subscales varied from 0.64 to 0.83, and the confirmatory factor analysis revealed a high structural validity. Subsequent work [34] reported high subscale reliabilities with moderate validity based on the scores from a large sample of ninth-grade students.

#### 2.2.6. Motivational Climate in Physical Education Scale (MCPES) 

The Finnish version of the MCPES [35] was used to measure the motivational climate in the soccer, ice hockey, and basketball sports clubs. The scale is composed of 18 items and four subscales: task-involving climate, ego-involving climate, social relatedness, and autonomy support. We modified the MCPES to reflect the context of the analyzed sports. The participants were asked to consider the stem, “On this team…”, and then to rate items on a 5-point Likert scale ranging from 1 (strongly disagree) to 5 (strongly agree). Concerning the psychometric properties of the MCPES, Soini and his colleagues [35] extensively examined the MCPES with 2594 girls and 1803 boys in the ninth grade. The researchers reported a satisfactory statistical fit for the four-factor model and acceptable Cronbach alpha [29] coefficients ranging from 0.78 to 0.88 based on the scored responses. 

#### 2.2.7. Descriptive Information

In the T1 data collection, the youth self-reported their gender, height, and weight. In the T2 data collection, participants self-reported by checking off their highest level of competition based on provided examples.

### 2.3. Procedure

The national soccer, ice hockey, and basketball federations provided the information related to the participants’ playing license, which contained the athlete’s participation status. In 2010, participants born in 1995 and with an active participation status (*n* = 9970) were invited to participate in the study. They received an envelope including information about the study’s purpose, a multisection questionnaire, instructions on how to complete the questionnaires, a statement that ensured the confidentiality of the participants’ responses, and a prepaid envelope to facilitate respondents’ response. Additionally, parental approval was requested from all the participants included in the study. Participation in the study was voluntary, and the participants had the option to withdraw from the study at any time without providing any reason for their decision. Participants and the sports federations gave informed consent, and the University of Jyväskylä provided ethical approval (488/1999). The overall response rate for the first data collection was 22.20% (*n* = 2014). The second data collection followed the same approach as the first data collection. Those (*n* = 2014) who answered the questionnaires in 2010 were contacted by mail and invited to participate in the second data collection. They received an envelope including information about the study’s purpose, a multisection questionnaire, instructions on how to complete the questionnaires, a prepaid envelope to facilitate respondents’ response, and a statement that ensured the confidentiality of the participants’ responses. The overall response rate for the second data collection was 41.75% (*n* = 841). Due to missing data, 17 participants were excluded from the statistical analysis.

### 2.4. Statistical Analyses

First, the participant characteristics (i.e., height, weight, and birth date) were examined, as well as the overall pattern of variable relationships. To examine our main purpose, the differences in individual and climate perceptions as well as sport enjoyment and perceived competence between elite and nonelite athletes, multivariate analysis of variance (MANOVA) for groups of variables (e.g., task and ego goal orientations) or analysis of variance (ANOVA) for singular constructs (e.g., perceived competence, sport enjoyment) was used. For the overall MANOVA or ANOVA test statistics, the partial eta squared (*η*^2^) statistic was also examined for meaningfulness, with interpretations being 0.01 as small, 0.06 as moderate, and 0.14 as large [36]. Based on the initial MANOVA or ANOVA results, the logical cascade of follow-up test statistics were examined. All significant (*p* < 0.05) psychosocial univariate F-tests were followed up with effect size calculations (Hedges’ *g*) calculated via https://www.polyu.edu.hk/mm/effectsizefaqs/calculator/calculator.html. We followed Cohen’s [37] interpretation guidelines for effect sizes. Hedge’s *g* of 0.20 was considered small, 0.50 medium, 0.80 large, and 1.20 or greater was considered very large. Except for Hedges’ *g*, statistical analysis was conducted using IBM SPSS Statistics version 24.

## 3. Results

### 3.1. Overall Pattern of Individual Motivations, Achievement Goal Climate Perceptions, Perceived Physical Competence, and Sport Enjoyment

Table 1 contains descriptive data and the intercorrelations among T1 and T2 study variables. Consistent with expectations and the youth literature, the participants perceived their climate to be more task- than ego-oriented, were higher in task than ego orientation, enjoyed their sports experiences, felt more autonomous than controlled, and were low in amotivation. As with the mean data, the correlations amongst the variables were theoretically consistent. The variables were relatively nonoverlapping, with only one correlation (autonomous motivation and task orientation) reaching 0.50. Last, all reliability coefficients were acceptable [29], with all being equal to or greater than 0.74.

### 3.2. T1 Results

Table 2 contains the descriptive data for the significantly different T1 and T2 variables between the elite and nonelite participants. Regarding motivational climate, the MANOVA group main effect was not statistically significant: F (2, 817) = 2.32, *p* > 0.05, Wilks’ Λ = 0.99. However, the task climate univariate F-tests were statistically significant, with the T1 elite athletes perceiving a greater task climate (3.91 ± 0.59) than the nonelite athletes: (3.76 ± 0.56), F(1, 818) = 4.66, *p* < 0.05. The MANOVA concerning goal orientations was not statistically significant: F(2, 816) = 1.86, *p* > 0.05, Wilks’Λ = 0.99. The MANOVA results relative to the two groupings of athletes’ motivations (autonomous, controlled, and amotivation) was significant: F(3, 806) = 8.14, *p* < .01, Wilks’Λ = 0.97, η^2^ = 0.09. Two of the univariate analyses were significant. The elite athletes were higher in autonomous motivation than the nonelite athletes (F(1, 808) = 11.37 *p* < 0.01) and lower in amotivation (F(1, 808) = 10.23, *p* < 0.01). Concerning our two singular constructs, the eventually elite athletes rated their perceived physical competence (F(1, 818) = 5.30, *p* < 0.05) and sport enjoyment (F(1, 546) = 11.33, *p* < 0.01) higher than the group of eventually nonelite athletes.

### 3.3. T2 Results

The MANOVA concerning perceptions of motivational climate indicated a statistically significant group main effect: F(4, 809) = 11.55, *p* < 0.01, Wilks’ Λ = 0.94, η^2^ = 0.03. The univariate analyses showed that elite and nonelite athletes differed regarding task climate: F(1, 812) = 41.43, *p* < 0.01. Likewise, elite and nonelite athletes also differed in social relatedness: F(1, 812) = 13.95, *p* < 0.01.

## 4. Discussion

Although children in Finland participate in high numbers in the Finnish youth sports club system, consistent success at the elite levels (e.g., the Olympic Games, World Championship, and professional teams associated with EuroLeague Championship events) is an ongoing national concern. Therefore, our study purpose was gaining a better understanding of the elite athletic development in Finland from the two dominant motivation frameworks in sports psychology to help guide and improve the sports club system. Our study followed youth who voluntarily self-responded to our questionnaires at two time points over the course of four years. We examined at the following time points: T1 (prospectively) and T2 (retrospectively). At T2, the participants self-reported becoming an elite athlete. At T1 data collection, none of the participants were participating as elite world-class or professional-level athletes in soccer, ice hockey, or basketball.

Prospectively and retrospectively, the results indicated that the eventually self-identified elite athletes compared with the nonelite athletes reported perceiving a significantly higher perception of a task climate. The importance of a task climate as related to our other measured variables is evident in the meta-analysis of motivational climate correlates by Harwood et al. [38]. For instance, in their meta-analysis, the task climate correlation with intrinsic motivation and overall motivation indices, such as relative autonomy, was large. Additionally, correlations with variables that relate to our measures of perceived physical competence and sport enjoyment, and the task climate correlations with confidence, self-esteem, and positive affect were moderate in magnitude. Although our work and Harwood and colleagues’ meta-analysis is not causal, a task climate appears, beside talent level and resources available, to underlie one possible pathway to becoming an elite performer. Also, concerning climate perceptions, the elite athletes retrospectively indicated perceiving greater levels of social relatedness than nonelite athletes. Evidence in the literature corroborates with the findings, indicating that social support plays a vital role in talent development [5], although Sarmento and colleagues [6] concluded that social influences are understudied regarding the talent development process.

Along the lines of a task climate, it was somewhat surprising that the elite and nonelite participants’ task orientation scores did not significantly differ at T1. Although the meta-analysis conducted by Lochbaum and colleagues [19] concluded only some support for elite athletes being higher in task orientation than nonelite athletes, others have reported differences in task orientation across different levels of elite performance. For instance, Höner and Feichtinger [39] studied youth soccer athletes across a four-year period, as in this investigation; however, from a younger age group, under 12 years (U12), to progression to U16. The task orientation was a significant psychological construct for cross-sectional data analyses at the U12 level, and then prospectively for selection in the professional clubs’ youth academies at the U16 level.

Concerning motivations, in youth field hockey athletes subdivided into elite and subelite categories, research [40] supports the overall notion that more elite youth are more motivated than subelite players. This result seemingly is in line with our T1 autonomous motivation and amotivation differences. Fenton et al. [41] demonstrated that more autonomously motivated youth soccer players engage in a higher percentage of time of moderate to vigorous physical activity during sport play. Hence, our prospective results combined with past research point to the importance of overall motivation that may lead to greater effort and involvement during sports club participation time.

With regards to our two singular constructs, at T1, the elite athletes self-reported higher levels of perceived physical competence and sport enjoyment compared to nonelite athletes. Systematic research [5,6,23] strongly supports the importance of both perceived competence and sport enjoyment as being interconnected and determinants of talent development and continuation in youth sports. Additionally, as previously discussed, perceived competence and sport enjoyment are integral in both OIT (and SDT overall) and AGT, as well as being related to autonomous motivation and a task climate. Our results suggested that the pathway to elite performance requires higher perceptions of both constructs during adolescence. Sports clubs should strive to promote both competence and sport enjoyment.

### Limitations and Future Directions

Despite the large sample and prospective nature of most of the data analyzed, this study has some limitations. One limitation concerns the methodological approach; more specifically, the employed scales. The use of a different climate scale and fewer scales retrospectively did not allow us to investigate the progression of any one motivational variable more than task climate. Hence, we encourage researchers to employ the same scales with multiple-timepoint studies. The next limitation concerned the ability to fact-check all responses concerning the attainment of elite status. Designs such as those employed by Höner and Feichtinger [39] (i.e., with the ability to follow youth from U12 to U16 academy selection) allow for a check of relationships among psychosocial variables and upward performance attainment. Working within one sport, as did Höner and Feichtinger, although the results might not be generalizable, most likely has the advantage of following all participants more closely. However, the number of participants in our study who indicated elite status was low, and thus we were not concerned with false reporting. It could also be because of the voluntary nature of participation that we missed out on elite athletes. Additionally, concerning elite status attainment, we did not measure the date and length of elite status attainment. We suggest that future investigations gain a fuller understanding of the athletes’ sporting careers. Last, although the sample size was seemingly large, from the initial invitations sent in 2010, only 8.43% of the eligible participants in 2010 were included in this investigation. 

## 5. Conclusions

In this study, we investigated individual and climate perceptions of Finnish youth athletes, as well as sport enjoyment and perceived physical competence. Based on the analyses of prospective and retrospective variables, Finnish youth athletes who self-identified as reaching elite athlete status reported higher perceptions of task climate, prospectively and retrospectively, than the nonelite status participants, thus strongly supporting the importance of a task climate within all levels of club sports. Additionally, participants who identified themselves as obtaining elite status reported prospectively higher levels of perceived competence, enjoyment, autonomous motivation, and a lower level of amotivation compared to nonelite athletes. As discussed, all of these variables are meta-analytically related to a task climate, thus strongly supporting the need for sport clubs and their coaches to promote a task climate. Club sports keep score, and they always will. However, emphasizing personal improvement, working hard, and enjoying the process are reasonable ways to promote a task climate. Last, retrospectively, the elite sample reported social relatedness as being important to their sports club experience. Therefore, the Finnish sports club system will seemingly benefit by incorporating task climate coaching while making the young athletes feel connected, to develop their youth into world-class athletes.

## Figures and Tables

**Table 1 sports-06-00165-t001:** Descriptive statistics and correlation among the study variables.

	M	SD	α	1	2	3	4	5	6	7	8	9	10	11	12
T1															
1. TC	3.78	0.59	0.89	-											
2. EC	2.50	0.70	0.83	**−0.13**	-										
3. TO	4.19	0.63	0.83	**0.43**	0.01	-									
4. EO	3.36	0.90	0.91	**0.13**	**0.34**	**0.24**	-								
5. PPC	37.19	4.73	0.74	**0.21**	0.03	**0.25**	**0.23**	-							
6. SE	4.59	0.62	0.93	**0.39**	−0.03	**0.30**	**0.14**	**0.28**	-						
7. AM	3.45	0.69	0.91	**0.39**	**0.18**	**0.50**	**0.22**	**0.22**	**0.41**	-					
8. CM	3.15	0.74	0.81	**0.22**	**0.25**	**0.19**	**0.29**	0.06	**0.12**	**0.56**	-				
9. Am	1.92	0.88	0.79	**−0.24**	**0.27**	**−0.19**	0.00	**−0.22**	**−0.48**	−0.07	**0.21**	-			
T2															
10. TC	3.80	0.74	0.83	**0.35**	0.02	**0.18**	0.03	**0.17**	**0.25**	**0.23**	0.07	**−0.17**	-		
11. EC	2.77	0.91	0.91	0.04	**0.33**	0.04	**0.17**	−0.01	0.01	**0.13**	**0.11**	0.08	**0.12**	-	
12. SR	3.50	0.97	0.97	**0.23**	−0.07	0.07	−0.02	**0.11**	**0.16**	**0.09**	0.04	**−0.14**	**0.52**	**−0.18**	-
13. AS	2.73	0.84	0.84	0.04	0.02	0.01	0.06	−0.02	0.02	0.05	**0.09**	0.08	**0.09**	**0.09**	**0.23**

Notes: TC = task climate. EC = ego climate. TO = task orientation. EO = ego orientation. PPC = perceived physical competence. SE = sport enjoyment. AM = autonomous motivation. CM = controlled motivation. Am = amotivation. SR = social relatedness. AS = autonomy support. Correlation is significant at the 0.01 level (1-tailed).

**Table 2 sports-06-00165-t002:** Means (*M*), standard deviations (*SD*), and Hedges’ *g* effect size (*g*) for elite and nonelite athletes’ variables that were significantly different (*p* < 0.05).

	Elite		Nonelite			
Time and Variables	*M*	*SD*	*M*	*SD*	*g*	*g* Interpretation
T1						
Task Climate	3.91	0.59	3.76	0.56	0.25	Small
Perceived Competence	38.35	3.85	37.06	4.82	0.27	Small
Sport Enjoyment	4.85	0.30	4.56	0.63	0.47	Nearly moderate
Autonomous Motivation	3.70	0.61	3.42	0.69	0.39	Small
Amotivation	1.61	0.70	1.94	0.89	−0.37	Small
T2						
Task Climate	4.31	0.55	3.75	0.74	0.76	Nearly large
Social Relatedness	3.88	0.88	3.46	0.96	0.44	Nearly moderate

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
