# Peer review of "Individual Motivations, Motivational Climate, Enjoyment, and Physical Competence Perceptions in Finnish Team Sport Athletes: A Prospective and Retrospective Study"

_sports, 2018, doi:10.3390/sports6040165_

Round 1
Reviewer 1 Report
Please include in introduction another or recent studies in this topic.
Please add and others parameters to table 1 because looks too simple and point after the title of tables 1,2 and 3.
The references are not in line to the policy of the journal (see abbreviations of journals).
Author Response
Review comments addressed in attached file.

Reviewer 2 Report
I would like to congratulate the authors on the presentation of a well written and highly interesting manuscript. The manuscript provides a detailed analysis of the motivational climates experienced by youth athletes and reports some key significant differences for those making elite levels of performance.
Broad Comments -
The introduction is nicely constructed and provides detailed background information on the topic and the theoretical framework. The methods are clearly explained and logical. This was a comprehensive study, which I again applaud the authors for. The results are well presented and clear. Logical conclusions are made throughout the discussions. The findings can be easily applied to a number of sports and (I would imagine) are relevant to most other countries with similar sport/athlete development systems. The suggestion of incorporating task climate coaching within the club system is important, if anything the authors could provide some practical suggestions as to how this could be achieved.
Specific Comments -
I feel the article title could be altered to provide a better understanding of the study. For me "Individual and Climate Perceptions" is awkward and would be better with a clear link to motivation / motivational climates.
Line 12 - Abstract "Despite the high rates of sports club participation among Finnish youth" is awkwardly worded. Could be revised to "Despite the high rates of participation in sports clubs among Finnish youth".
Line 33 - insert "of" before becoming elite athletes.
2.2 Measures - can validity and reliability scores of the questionnaires be included in this section.
Line 252 - "participant" needs to be revised to "participate"
Author Response
Comments addressed in attached file.

Reviewer 3 Report
Page 1 – Title – should it state something about which climate – motivational climate?
Abstract:
Page 1 – line 13: Perceived motivational climate? The authors seem to use different terms? Please use the same.
Page 1 – line 13: sport enjoyment instead of enjoyment? See the rest of the text
Page 1 – line 17: motivational variables: please specify
Page 1 – line 21: significantly higher differences?
Page 1 – line 28: sport enjoyment
Introduction
Page 2 – line 49: please specify approximately number of athletes?
Page 2 – line 65: coaches´ support boost autonomous.. could this be rephrased
Page 2 – line 91-93: please rephrase would self-reported higher…
Page 2 – line 95: Types; task ..
Materials and Methods
Page 3 – line 100: you could use T1 instead of Time 1. Etc.
Page 3 – line 104: is the somewhat skewness in terms of the overrepresentations in soccer discussed in the article? Should it? Potential explanation/bias on some of the results?
Page 3 – line 104: the same with gender? Elite athletes Time 2.
Page 4 – line 146: is this measurement related to Physical Activity Enjoyment Scale (PAES)?
Page 4 – line 155: Why was MCPES used in stead of PMCSQ used in Time 1?? What is the logic? Physical Education?
Page 4 – line 184: so the overall response rate was 8,43 %?
Results
page 5 – line 210: but you could have used the data from the sample compared to the elite?
Page 6 – line 225: Time 1 and Time 2 data? Both are included in this section?
Page 6 – line 237: perceived confidence, which measurement is that based on?
Page 6 – line 239; the table, there are some numbers in bold letters, what are they? The significant ones are marked with a *?
Page 7 – line 249: the Hedges´g values could have been listed in the table?
page 7 – line 261-3: see the point, but what is the relevance for the article?
Author Response
Comments addressed in attached file.
